# COVID-WAREHOUSE: A Data Warehouse of Italian COVID-19, Pollution, and Climate Data

**DOI:** 10.3390/ijerph17155596

**Published:** 2020-08-03

**Authors:** Giuseppe Agapito, Chiara Zucco, Mario Cannataro

**Affiliations:** 1Department of Legal, Economic and Social Sciences, University Magna Graecia of Catanzaro, 88100 Catanzaro, Italy; agapito@unicz.it; 2Data Analytics Research Center, University Magna Graecia of Catanzaro, 88100 Catanzaro, Italy; 3Department of Medical and Surgical Sciences, University Magna Graecia of Catanzaro, 88100 Catanzaro, Italy; chiara.zucco@studenti.unicz.it

**Keywords:** Italian COVID-19 data, data analysis, data warehouse, data integration, pollution data, climate data

## Abstract

The management of the COVID-19 pandemic presents several unprecedented challenges in different fields, from medicine to biology, from public health to social science, that may benefit from computing methods able to integrate the increasing available COVID-19 and related data (e.g., pollution, demographics, climate, etc.). With the aim to face the COVID-19 data collection, harmonization and integration problems, we present the design and development of COVID-WAREHOUSE, a data warehouse that models, integrates and stores the COVID-19 data made available daily by the Italian Protezione Civile Department and several pollution and climate data made available by the Italian Regions. After an automatic ETL (Extraction, Transformation and Loading) step, COVID-19 cases, pollution measures and climate data, are integrated and organized using the Dimensional Fact Model, using two main dimensions: time and geographical location. COVID-WAREHOUSE supports OLAP (On-Line Analytical Processing) analysis, provides a heatmap visualizer, and allows easy extraction of selected data for further analysis. The proposed tool can be used in the context of Public Health to underline how the pandemic is spreading, with respect to time and geographical location, and to correlate the pandemic to pollution and climate data in a specific region. Moreover, public decision-makers could use the tool to discover combinations of pollution and climate conditions correlated to an increase of the pandemic, and thus, they could act in a consequent manner. Case studies based on data cubes built on data from Lombardia and Puglia regions are discussed. Our preliminary findings indicate that COVID-19 pandemic is significantly spread in regions characterized by high concentration of particulate in the air and the absence of rain and wind, as even stated in other works available in literature.

## 1. Introduction

The COVID-19 (COronaVIrus Disease 2019) outbreak is caused by a novel coronavirus named Severe Acute Respiratory Syndrome CoronaVirus 2 (SARS-CoV-2) [1] and has been classified as a pandemic disease by the World Health Organization (WHO) on 12 March 2020. SARS-CoV-2 has spread over all the world in less than six months, causing more than 10 million tested-positive cases and more than half a million confirmed deaths [2].

A global response has been quickly developed in the form of collective data collection and analysis efforts, which are generally aimed to understand SARS-CoV-2 biology and delivering therapeutic solutions in clinical/pharmacological protocols [3].

Thus, the management of the COVID-19 pandemic presents several unprecedented challenges that regard a plurality of fields and that may benefit from computing infrastructures and software pipelines, with the main aim to integrate the increasing COVID-19 publicly available data to allow their full exploitation and world-wide collaboration.

COVID-19 poses many challenges to several research and application fields that regard, to cite a few: molecular basis of the disease, virus mutations, vaccines and drugs, diagnosis and therapy, ICUs (Intensive Care Units) management [4], healthcare logistic [5], large scale testing of people to find diseased people and already healed people, large scale tracing of people movements and contacts to reduce the spread of the virus, infectious disease modeling [6], epidemiology, public health, effects of pandemic at emotional and behaviour level [7], impact of pandemic on remote working [8], etc.

Each one of these challenges may benefit from advanced computing infrastructures and novel software pipelines [9], here we focus on the issue of data integration of publicly available COVID-19 data, that may simplify data visualization and aggregation, e.g., for decision making and for focusing on specific aspects of the problem, or may simplify the connection of such disease data with environmental and climate data [10,11].

A main issue of current publicly accessible COVID-19 data, is that it is provided in raw textual format, such as Comma Separated Value (CSV) files. On one hand, this allows an easy download of data, but on the other hand, data is not structured and requires heavy pre-processing and ingestion activities for further analysis.

Main current COVID-19 data is provided by government agencies. Focusing on the Italian *Protezione Civile* Department, data is communicated daily by the *national* Protezione Civile branch, and contains a set of measures detected daily on the territory, such as the number of daily infected subjects quarantined at home, hospitalized patients, hospitalized patients into ICU, deceased subjects, healed subjects, performed swab, positive and negative swabs, and so on. Such measures are roughly geo-localized to local administrative entities, such as provinces, regions and the overall nation. Moreover, *local* Protezione Civile branches provide the regional data that is collected by the national Protezione Civile branch. Finally, data is provided through collections of CSV files organized in provinces, regions, nation. This data may be dowloaded at the following URL: https://github.com/pcm-dpc/COVID-19.

Although the data spread in such various CSV files is provided in raw unstructured format, it can be conceptually organized using a classical data warehouse (DW) data model, e.g., using the Dimensional Fact Model. The Dimensional Fact Model uses three main concepts: *facts*, that refer to the subject of study (e.g., the study of deceased people due to COVID-19), *measures* that refer to the measured data about the studied phenomena (e.g., the number of deceased people due to COVID-19, in a given time and place), and *dimensions* that refer to hierarchies along which measures can be considered and aggregated (e.g., the temporal dimension may include the day, week, month, year hierarchy, while the geographical dimension may include the city, province, region, nation hierarchy). On each dimension, several aggregation functions can be defined, such as the sum, median, medium value of all measures data points (e.g., the deceased people in March 2020 in the Calabria region).

In summary, relevant data, named facts, are used to analyze a phenomena and are measured through some variables named measures. Measures are collected at different points along one or more dimensions. The dimensions can be considered as the filters that will allow us to group (and analyze) the data collected in the DW. Examples of dimensions are the geographical dimension, the temporal dimension, or type dimension. Here the measures are the measured COVID-19 data provided by the Italian Protezione Civile Department, while the dimensions are the temporal dimension and the geographical dimension.

The advantage to model COVID-19 data using the DW model is the possibility to build relevant data cubes on the top of the DW and to apply well known OLAP (On Line Analytics Processing) analysis operations.

For instance, considering the data cube of deceased patients (the fact), that contains all the measured number of deceased people per day and per location, by considering the temporal and geographical dimensions, the OLAP model provides several analysis operations, such as Slice and Dice or Drill-up and Drill-down.

The main contribution of this paper is the design and development of a data warehouse named COVID-WAREHOUSE that models and stores the COVID-19 data made available daily by the Italian Protezione Civile Department using the Dimensional Fact Model. A second contribution is the integration in such data warehouse of some pollution and climate data made available by the Lombardia and Puglia regions. COVID-WAREHOUSE allows to aggregate and view data using standard OLAP functions and allows to study the possible correlation between COVID-19 data and pollution and climate data in Italy. The rest of the paper is organized as follows. Section 2 recalls recent works aimed to build centralized repositories of COVID data, that jointly analyzed COVID-19 and environmental data. Section 3 describes the COVID-19, pollution and climate data that have been integrated into the data warehouse. Section 4 presents the design and implementation of COVID-WAREHOUSE. Section 5 presents some case study showing relevant data cubes built on COVID-WAREHOUSE and different data analysis performed on them. Moreover, some correlations among COVID-19 and environmental data are presented. Section 6 concludes the article and outlines future work.

## 2. Background

In this Section, we first report some recent initiatives aiming to build data warehouses of COVID-19 data, along with some open-source initiatives to build data warehouse systems. Afterward, we present some recent works that integrated and analyzed COVID-19 data with environmental and pollution data.

### 2.1. Data Warehouse Systems

Here, we summarize some of the most used open-source data warehouse systems.

**BIRT** (http://www.eclipse.org/birt/) is an open-source software project that provides the technologies and platform to create data visualizations and reports that can be embedded in rich client and web applications. Because BIRT is developed by using Java technology. BIRT is a top-level software project within the Eclipse Foundation, an independent not-for-profit consortium of software industry vendors and an open-source community. BIRT is distributed under the Eclipse Public License (EPL). BIRT comprises two main components: *(i)* a visual report designer for creating BIRT Designs, and *(ii)* a runtime component for generating designs that can be deployed on any Java environment. The BIRT project also includes a charting engine that is both fully integrated into the BIRT designer and can be used standalone to incorporate charts into an application. BIRT designs are made persistent as XML and can access several different data sources, including JDO datastores, JFire Scripting Objects, POJOs, SQL databases, Web Services, and XML.

**OpenReports** is a powerful and easy-to-use open-source web-based reporting solution that provides dynamic report generation capabilities. OpenReports supports a variety of open-source reporting engines, including JasperReports, JFreeReport, JXLS, and Eclipse BIRT. It also includes QueryReports and ChartReports, to create SQL based reports that do not require a predefined report definition. Additionally, OpenReports now supports OLAP, via Mondrian and JPivot. OpenReports provides a web-based report generation and administration interface with the following features: support for a wide variety of export formats, such as PDF, HTML, CSV, XLS, RTF, and Image; support for Drill-Down reports and external application integration via secure report generation URL. OpenReports reports generation and scheduling capabilities are also available directly via the ReportService, a Service-oriented architecture (SOA) for report generation, through a comprehensive and flexible API exposed as a SOAP web service and to HTTP GET/PUT requests. OpenReports’ source code can be downloaded from the SourceForge project page (http://sourceforge.net/projects/oreports/).

**Intermine** (http://intermine.org) is a powerful open-source data warehouse system [12]; it allows users to integrate diverse data sources with minimal effort, providing powerful web-services and an elegant web-application with minimal configuration. InterMine is an open-source data warehouse released under the license (LGPL 2.1) and free to use explicitly built for the integration and analysis of complex biological data. It powers some of the biggest data warehouses in the life sciences, including FlyMine, HumanMine, MouseMine, YeastMine, ZebrafishMine, RatMine, TargetMine, ThaleMine, and PhytoMine. Intermine provides dynamic data tables that allow users to easily drill down into data, filter data, add additional columns, and navigate to report pages. More details about the full functionalities are available on Intermine website (https://intermine.readthedocs.io/en/latest/).

The article available here (https://www.stitchdata.com/integrations/covid-19/) discusses one of the issues faced by our paper, i.e., the automatic ingestion of publicly available COVID-19 CSV files into a structured data warehouse. Starting from the consideration that the main public available COVID-19 datasets are CSV files, the proposed Stitch integration tool supports the Extraction Transformation and Loading (ETL) of such COVID-19 datasets into the user data warehouse. The Stitch integration is a set of guidelines and tools to build a data warehouse that requires several steps (https://www.stitchdata.com/docs/integrations/saas/covid-19). Although the system seems to cover several data sources stored in GitHub (for which the system provides metadata about each data source schema), its usage is free for seven days only. This tool has been created by a collaboration between the Singer open-source community, Talend and Bytecode (https://www.talend.com/blog/2020/04/05/talend-joins-fight-against-covid-19-unlocking-data-for-health-researchers/). The COVID-19 integration includes the following datasets:Johns Hopkins CSSE Data (https://github.com/CSSEGISandData/COVID-19), that contains COVID-19 data provided by JHU CSSE (https://systems.jhu.edu/research/public-health/ncov/);EU Data (https://github.com/covid19-eu-zh/covid19-eu-data), that contains automated data collection of COVID-19/SARS-COV-2 cases in EU by Country, State, Province, Local Authorities, and Date;Italy Data (https://github.com/pcm-dpc/COVID-19), that contains automated data collection of COVID-19/SARS-COV-2 cases in Italy by Regions, Province, and Date;NY Times U.S. Data (https://github.com/nytimes/covid-19-data), that contains data on coronavirus cases and deaths in the U.S. (https://www.nytimes.com/interactive/2020/us/coronavirus-us-cases.html);Neher Lab Scenarios Data (https://github.com/neherlab/covid19_scenarios_data), that contains data preprocessing scripts and preprocessed data for COVID-19 Scenarios project (https://covid19-scenarios.org/), a mathematical model simulating several COVID-19 outcomes on the basis of some user-defined parameters;COVID-19 Tracking Project (https://github.com/COVID19Tracking), that collects and publishes data for US states and territories (https://covidtracking.com/).

The *COVID-19/2019-nCoV Time Series Infection Data Warehouse* available here (https://github.com/BlankerL/DXY-COVID-19-Data/blob/master/README.en.md) contains the COVID-19 Global Pandemic Real-Time Report provided by Ding Xiang Yuan (https://ncov.dxy.cn/ncovh5/view/en_pneumonia). Ding Xiang Yuan (DXY) is a medical online social networking service for China’s physicians and medical professionals. The data is obtained by COVID-19 Infection Data Realtime Crawler. The data is published hourly and is provided as CSV files. As data sources, it uses both Chinese and International data sources.

Google had launched the *COVID-19 public dataset program* (https://cloud.google.com/blog/products/data-analytics/free-public-datasets-for-covid19). This repository contains public datasets, like Johns Hopkins Center for Systems Science and Engineering (JHU CSSE), the Global Health Data from the World Bank, and OpenStreetMap data, free to access and query (https://console.cloud.google.com/marketplace/details/bigquery-public-datasets/covid19-dataset-list\?preview=bigquery-public-datasets).

The article available here (https://towardsdatascience.com/a-short-review-of-covid-19-data-sources-ba7f7aa1c342) describes some further initiatives and in particular discusses the strength and weakness of the following data sources:1 Point 3 Acres (https://coronavirus.1point3acres.com/en);Johns Hopkins CSSE (https://github.com/CSSEGISandData/COVID-19);The COVID Tracking Project (https://covidtracking.com);The Kaggle Novel Coronavirus Dataset (https://www.kaggle.com/sudalairajkumar/novel-corona-virus-2019-dataset);Ding Xiang Yuan (https://ncov.dxy.cn/ncovh5/view/en_pneumonia);

In summary, all the COVID-19 repositories are unstructured collections of CSV files. Only STICH proposes a data warehouse of COVID-19 data, but it is not free nor open, so we did not use it. In addition, we reported some open-sources projects BIRT, OpenReports, and Intermine. BIRT provides the assets to create data visualizations and reports embedded in web applications. OpenReports provides the tools to create dynamic reports in web applications. Finally, Intermine is data warehouse system for the integration and analysis of complex biological data.

### 2.2. Methods to Integrate and Analyze Covid-19 Data with Environmental and Pollution Data

Here, we present some recent works devoted to integrate and analyze COVID-19 data with environmental and pollution data. In the work [13], authors correlated the data about deceased people in USA due to COVID-19 with pollution data and the authors demonstrated, using different statistical methods, that a very small increase in the fine particulate matter PM2.5 is responsible for a very great increase in mortality of COVID-19 patients.

In the work [14], authors used the Group Method of Data Handling (GMDH) neural network to analyze climate and confirmed COVID-19 case in the Hubei province in China, considering a 30 days period. Maximum, minimum, average daily temperature, relative humidity, wind speed, and density of population, were used as the input dataset for predicting the number of confirmed cases. The classification model was able to predict confirmed cases with good accuracy. Moreover, regression analysis demonstrated that relative humidity and maximum daily temperature had a strong impact on confirmed cases: relative humidity affected positively the confirmed cases, while maximum daily temperature affected negatively the confirmed cases. These results confirm other studies reported in [15].

In the work [16], the authors study the impact of climate and urban measures on confirmed cases of COVID-19 in three Italian regions using multivariate linear regression (MLR). The MLR model demonstrated that some climate (average temperature, relative humidity, wind speed) and urban measures (population density) impact on confirmed cases of COVID-19.

The works presented so far do not offer any integrated curated repository of those data.

## 3. Italian Covid-19, Pollution and Climate Data

Regional differences related to the cumulative incidence and its variation in the number of cases, deaths, and complications related to COVID-19, could prove useful in identifying a relationship between epidemiological and environmental factors. To obtain a more in-depth overview, it is necessary to integrate adequate data and information for analysis. In this paper, three main categories of data have been considered:Epidemiological COVID-19 data,Air pollution data,Weather data.

The current Section provides an overview of each of the aforementioned categories of data available for Italy at different administrative division levels.

### 3.1. Epidemiological Covid-19 Data

Since 24/02/2020, a daily updated repository containing the official data on Italian COVID-19 has been made publicly available by the Italian Department of Civil Protection at https://github.com/pcm-dpc/COVID-19.

The provided repository contains information related to the number of total cases, current, and daily new cases, as well as the number of Hospitalized, Recovered, and Death patients. Data is provided in a time-series data fashion, and collected at three different administrative division levels: National level, Regional or first-level, and Provincial or second-level administrative division. For each division, different information is reported in a comma-separated values (CSV) file. In particular, daily data provided at regional levels are collected from all the nineteen Italian regions and the 2 autonomous provinces of Trento and Bolzano. Regional data for each field is aggregated and stored at a national level, while the Provincial dataset only reports information for the total number of cases per Province. This heavily limits the possibility of making analysis at a province level. Some Regions provide province data but often in an unstructured way, e.g., in *HTML* format. The available data fields are: the used Date format (e.g., YYYY-MM- DDTHH:MM:SS (CEST)), the Country Reference (e.g., XYZ ISO 3166-1 alpha-3), Region Id Number, Region Name, Province Id Number, Province Name, Hospitalised patients (not ICU), Patients in ICU, Total hospitalised patients (e.g, hospitalised patient plus patient in ICU), Number of home-treated patients, Total amount of Positive Cases (e.g., the cumulative measure of Total hospitalised patients and Home-treated patients), Variation in daily incidence (e.g., the difference of the total amount of Positive Cases at current day and the total amount of Positive Cases at previous day), Variation in cumulative incidence (e.g., the difference of total cases at current day minus the total cases at previous day), Recovered, Death, Total amount of positive cases, Number of performed tests (SWABS), Number of tested people 24, Notes e.g., issues in reporting collection for that day, both in Italian and English language.

### 3.2. Air Pollution Data

Many scientific works are also counting air pollution among the main responsible for the spreading of the COVID-19 virus [13]. A crucial public health objective is to identify critical environmental factors, such as air pollution, that could boost the severity of the health outcomes (e.g., ICU hospitalization and death) among people affected by COVID-19. As stated in the work of Wu et al. [13], authors have determined that there is a substantial overlap between causes of deaths of COVID-19 patients and the diseased people that are affected by long-term exposure to fine particulate matter (PM2.5). In this regard, we collected from the Lombardia and Puglia Italian Regions (At the time of writing we were able to collect data only from those two regions.) the air pollution data. From the Lombardia region, it has been possible to download the pollution data for all the provinces, available at the ARPALombardia web site (https://www.arpalombardia.it/Pages/Aria/Richiesta-Dati.aspx#). The Lombardia air pollution data are stored in multiple CSV files arranged as couples Provincia and particulate, and sorted by detection data. The measured particulate particles enclose Nitrogen dioxide (NO${}_{2}$), nitric oxide (NOX), Carbon monoxide (CO), Sulfur dioxide (SO${}_{2}$), Ozone (O${}_{3}$), Ammonia (NH${}_{3}$), Benzene (C${}_{6}$H${}_{6}$), Particulate Matter (PM10), i.e., particulate matter with a diameter smaller than 10 micrometers, and Particulate Matter (PM2.5), i.e., particulate matter with a diameter of less than 2.5 micrometers. The detected concentration for each kind of particles in the air is expressed as μg/m${}^{3}$. An example of air pollution data available at the Lombardia Region web site is shown in Table 1.

From the Puglia region, we were able to download the pollution data for all the provinces, available at the ARPAPuglia web site (http://www.arpa.puglia.it/pentaho/ViewAction?solution=ARPAPUGLIA&path=metacatalogo&action=meta-aria.xaction). The Puglia air pollution data are stored either in multiple CSV files or in a single CSV file and are arranged as tuples. The attributes are the Province name, date and time of the measurement, the measured particulate values, the name of the detection unit, and the name of the particulate.

The measured particulate particles enclose Nitrogen dioxide (NO${}_{2}$), nitric oxide (NOX), Carbon monoxide (CO), Sulfur dioxide (SO2), Ozone (O3), Ammonia (NH${}_{3}$), Benzene (C${}_{6}$H${}_{6}$), hydrogen sulfide (H2S), impact pathways approach (IPA TOT), BLACK CARB, Particulate Matter (PM10), and Particulate Matter (PM2.5). The detected concentration for each kind of particles in the air is expressed as μg/m${}^{3}$. An example of air pollution data available at the Puglia Region web site is shown in Table 2.

Pollution data from Lombardia and Puglia have been downloaded manually.

### 3.3. Weather Data

Several are the studies investigating the relations among human coronaviruses and environmental humidity and temperature measures [17,18,19] In particular, recently proposed works aiming at investigating the effects of different meteorological conditions on the spread of COVID-19, may suggest an existing relationship between average temperature and COVID-19 incidence rates [20,21,22,23]. However, research in this field is still limited and, therefore, efforts are needed to provide reliable data sources. Concerning this, weather-related data has been automatically collected for every Italian province, from 2020/02/24, up to the date of writing, through automatic web scraping techniques.

For data collection, Python Beautiful Soap and requests libraries have been used to automatically collect weather information from the Italian website ”IlMeteo.it” (https://www.ilmeteo.it/portale/archivio-meteo/) and to store the data in a single CSV file. A pipeline for data collection is reported in Figure 1. For instance, to collect the weather data relating to the province of Milan on 24 March 2020, the Python code:reconstructs the URL relating to the weather archive page containing the weather information for the province of Milan on the desired date,connects to the page and, by parsing the HTML document,extracts the desired information.

The described process is then iterated through all the Italian provinces and all the dates that appear in the epidemiological data file. The information is stored in a *CSV* format file that is further ingested in the Data Warehouse.

The CSV stores data sorted by chronological order. The collected meteorological data include daily temperature minimum (°C), temperature maximum (°C), temperature average (°C), humidity maximum (%), humidity minimum (%), humidity average (%), wind speed average (km/h), higher sea level pressure (mb), the amount of Rainfall (mm) other weather phenomena.

## 4. Covid-Warehouse: Design and Implementation

The main features of the Data Warehouse (DW) are data integration and consistency, because the DW relies on multiple sources of heterogeneous data, i.e., data extracted from several internal or external information systems, or collected in various heterogeneous files. DW undertakes to return a unified vision of different data, generally represented in multidimensional form.

The basic idea is to see the data as points in a space whose dimensions correspond to as many possible dimensions of analysis; for example, a point could represent an event that occurred in an Italian Province, and it is described through a set of measures of interest, e.g., the average number of deceased people due to COVID-19 in that province. Traditional DW systems are designed to analyze a massive amount of historical data obtained from transactional data but can also include data from other sources. A general DW populating process encompasses the following steps: *(i)* Data Extraction from a database, or loading data from files to a database (*(i.a)* If loaded data are not normalised it is need to normalize them in Third Normal Form (3NF). 3NF is a methodology used to reduce the duplication of data, avoid data anomalies, and ensure referential integrity.); *(ii)* Data Transformation: it is necessary to denormalize and transform the database tables to yield summarized data, multidimensional views, and faster user response times of data analytics Dimensional Fact Model; *(iii)* Data Loading: tables are logical arranged in a multidimensional model known as snowflake, that centralizes the fact tables to a multidimensional schema (i.e., data cubes). The snowflake schema centralizes fact tables which are connected to multiple dimensions normalized into multiple related tables.

DW is developed to deal with an enormous volume of data efficiently (e.g., many years of collected data to support historical analysis) to be more effective in analytics performance. Thus, the strengths of the DW, are in the reconciliation and normalization of multiple data sources to homogenize and integrate the data and eliminate the inconsistencies, it allows to obtain a conceptual model for the data mart. The model adopted is known as the Dimensional Fact Model (DFM), used to describe all the concepts of the multidimensional model, i.e., facts, measures, dimensions, and hierarchies.

On the other hand, DW systems are ineffective to preform real time analysis dealing with a low volume of data in constant evolution, as the case of Italian COVID-19. The COVID-19 data are produced daily and in small quantities making the classical DW methodologies ineffective. Each day when new COVID data are available, it is necessary to update the previously collected data with the new ones. It is worthy to note, that COVID-19, air pollution and climate data are available as small CSV file. To analyze the daily available data, they have to be normalized (e.g., to remove missing, duplicated and useless values that might contribute to provide poor quality information) before to be stored in a databases. After normalization, to yield summarized data i.e., DFM, it is necessary denormalize and transform the database tables to obtain the new DFM, from which to obtain the snowflake form. Thus, daily data updates make classical DW ineffective both from a computational and performance point of view.

For these reasons, we developed COVID-WAREHOUSE, an automatic framework implemented in *Python*, coming with all the main features of the traditional DW system, and tailored to effectively works with small datasets.

The COVID-WAREHOUSE architecture encompasses 5 independent and cooperating levels: *(i)* data sources level; *(ii)* integration level; *(iii)* warehouse level; *(iv)* analysis level; and *(v)* data visualization level. Figure 2 shows the COVID-WAREHOUSE architecture.

The main functions of the 5 levels of COVID-WAREHOUSE architecture are described below.

***Data source level***: it is responsible for the data collection. The data are obtained from different external data repositories. Relevant COVID-19, and climatic data are automatically collected from the respective repositories by using automatic scripts written in Python and made available in COVID-WAREHOUSE. Whereas, pollution data are manually collected from the respective data repositories, and made available in COVID-WAREHOUSE.

***Integration level:*** it gets as input the collected data which must be extracted, and cleaned to eliminate inconsistencies and complete any missing parts. Finally, the normalized data can be integrated together according to a common schema. To make more effective this preprocessing phase, the ETL (Extraction, Transformation and Loading) approach, has been implemented using Pandas library [24], allowing to integrate heterogeneous schemes, as well as to extract, transform, clean up, validate, filter and load data from the CSV files and other data source in the COVID-WAREHOUSE. The main target of the preprocessing is to define the common schema to use to merge together the heterogeneous input data. In fact, to join together COVID-19 data with air pollution and climate data we must figure out if there exist common attributes or if they can be obtained by combining other available attributes. To merge together COVID-19 data with air pollution and climate data, we figure out the following attributes Date/Time, and Province name (i.e., that are available in all the input datasets) with which to create the joining schema. Before performing the merge, data have to be cleaned by removing duplicate values as well as removing comments and other annotations (for which, it was necessary a manual inspection of the input files). The chosen attributes for the join need to be normalized, otherwise it is impossible to obtain a match because data attributes are highly heterogeneous. In COVID-19 data the date format is: “2020-02-24 18:00:00”, in Lombardia air pollution, the date format is: “2020/01/01”, in Puglia air pollution, the date format is: “2020-02-24 00:00:00.0”, and finally in climate data the date format is: “2020/February/24”. We chose as common date format the following “2020-02-24”. Because the time of measurement is not relevant form the subsequent analysis we split the date in three new attributes: year, day and month and use them as dimensions in the cubes. The *Reconciled Table* (i.e., obtained joining all the input data) is implemented through the use of Pandas DataFrames. DataFrame is a 2-dimensional high-performance data structure suitable to represent a spreadsheet or SQL table. The *Reconciled Table* contains COVID data combined with air pollution and climatic data.

***Warehouse level:*** it is an information collector and plays the role of central and global container of summarized data, designed to enable business intelligence activities. It is developed to emulate the ability of DW to provide better analytical performance than transaction processing. We defined a central collector of information, to avoid a complex scheme of data accesses that can contribute to the risk of inconsistencies between the data marts, and to facilitate efficient storage, by enhancing the quality of real data analysis processes. Data Marts are data structures that are optimized for faster access. The DW is implemented using software components that can provide good performance to deal with both massive and limited amounts of data, to make better operational decisions. The DW module does not consist simply in a data container, but it also requires a customized architecture, able to collect, store, analyze, and present information. Thus, we implemented the DW as a data structure optimized to support quick and efficient access to multiple data sources, by tacking advantages from the Pandas *DataFrames*. Moreover, all the relevant data structures available in the DW system (e.g., DFM, snowflakes etc) used to store all the data are implemented by using DataFrames.

***Analysis level:*** it allows efficient and flexible consultation of integrated data for reporting, analysis and simulation purposes. From the Reconciled Table of the DW, the DFM is automatically created from which to yield the snowflake model, as well as the cubes (the aggregation of multidimensional data), focusing on specific facts of interest for decision making. A cube represents a set of data, which are described in principle by numerical measures. Each axis of the cube represents a possible dimension of analysis. Each dimension can be viewed at the most detailed levels identified by the attributes.

***Data Visualization:*** it allows to display in a graphical form all the data cubes obtained from the previous level. Data visualization allows to highlight in a remarkable way some features of the analyzed data that is impossible to figure out otherwise, helping researchers to improve data analysis process.

### The Normalization and Denormalization Approaches

Database and Data Warehouse optimization is an essential step to improve performance and provide better user experiences. To achieve better performance, the essential methods are Normalization and Denormalization respectively applied in databases and data warehouses. Normalization is a method of minimizing the number of insertion, deletion, and update due to the presence of redundant data that can produce anomalies in the database. The goal of the normalization process is to eliminate redundant data (e.g., the same data stored in more than one table), which are responsible for wasting disk space, and slow down system performance. Normalization involves the analysis of functional dependencies between attributes, allowing to distribute the data into multiple tables, reducing data redundancy and inconsistency to achieve data integrity. Normalization relies on the concepts of normal forms. A table is said to be in a normal form if it fulfills specific constraints. In particular, there are five normal forms 1NF, 2NF, 3NF, 4NF, and 5NF, plus the Boyce-Codd Normal Form (BCNF). Thus, Normalization is mainly applied in OnLine Transaction Processing (OLTP) systems, where it is mandatory to make insertion, update, and delete as fast as possible.

Denormalization is the opposite of Normalization because redundancy is added to the data to improve the performance of data warehouse applications while ensuring data integrity. The reason for performing Denormalization is to eliminate the overheads provided by the query processor in performing specific database queries that join data from many tables into one. Denormalization method highlights the concept that placing all the data in one place reduces the need to search multiple data structures to collect the searched data. Denormalization procedure regards the choice of attributes to add to existent tables that allow reducing the number of joins, improving the overall performance. Thus, Denormalization is applied in OnLine Analytical Processing (OLAP) systems, where it needs to analyze and retrieve as fast as possible historical data stored from multiple database systems at one time.

COVID-WAREHOUSE performance can be substantially improved by minimizing the number of accesses to secondary storage during transaction processing. The number of access to the secondary memory can be reduced by using Denormalization. Denormalization consists of reducing the number of physical tables necessary to be accessed to retrieve the desired data by reducing the number of required joins to derive the query. Denormalization in COVID-WAREHOUSE contributes to providing better performance as well as aggregated data in a form ready for immediate display.

## 5. Results

In this section, we present a case study to demonstrate the functionality, capability, and validity of the COVID-WAREHOUSE framework. All the analyses described in this section would have not been possible without the ability of COVID-WAREHOUSE to integrate Italian COVID-19 data to air pollution and climate data.

### 5.1. Preliminary Operations and Data Mart Building

The data mart we developed contains the Italian COVID-19, climate, and pollution data starting from 24 February 2020, up to the 31 March 2020. COVID-19 and climatic data are updated daily through the use of python scripts that periodically download new data, e.g., a script downloads Italian COVID-19 data every day after 6 p.m. Instead, the climatic data are downloaded every morning. Differently, air pollution data are collected each day manually. The latest data must be preprocessed to remove inconsistency and missing data before being annexed in the apposite COVID-WAREHOUSE tables. Often, downloaded data can contain information already in the data mart. In that case, the information from the latest data replaces the previous one. Thus, only the newest data are kept in the data mart. At the end of the preprocessing, data are automatically converted into suitable tables exploiting Pandas DataFrames [24]. In detail, the COVID-WAREHOUSE encloses several data frames, e.g., Italian province and region tables, air pollution, and climate tables, from which to create the data mart and the cubes quickly. The yielded cubes can be converted in reports easily, allowing the analyst to obtain a correlation between the presence of both PM10 and PM2.5 and the number of positive cases.

The main steps to create a multidimensional data cube from raw COVID-19, climatic, and pollution data are the following:Creating the data source by injecting or updating data into the COVID-WAREHOUSE data repository by using the available load update functions.Data Cleaning is done automatically by COVID-WAREHOUSE at the end of the first step. Raw data are automatically cleaned and encoded in a unique standard format using the ETL functions available in COVID-WAREHOUSE. Users should only give to the COVID-WAREHOUSE the name of the key attributes when required.Data Merging is performed automatically by COVID-WAREHOUSE at the end of the data cleaning process, yielding as output the Data Warehouse.Data Mart Creation can be done in two ways: *(i)* creating a new Reconciled table or *(ii)* selecting an existing Reconciled table to use as the Data Mart. The Reconciled table should contain the attributes, metrics, and other objects that users want to use as columns in the data mart table and populate the data mart. To create a new Data Mart, users must specify the attributes, and the metrics to create the data mart, and the related multidimensional cube.As the final step, COVID-WAREHOUSE automatically computes the correlation among the chosen attributes and metrics and visualizes the results as heatmap.

### 5.2. Data Analysis

Our analysis of Italian COVID-19 data enriched with climate, and air pollution data, pursues the objective to identify a possible correlation between climatic events (i.e., absence of rain, a windy day etc.), the measured level of pollution and the spread of the COVID-19.

The correlation is a statistical measure of the relationship among variables whose characteristic values belong to the range [−1,1], where values of correlation closer to −1 and 1 indicate a strong correlation between variables, while values nearest to 0 represent variables weakly related. In particular, there exist three types of correlation, that are: **Positive** where if a variable increases, the other one also increases, **Negative** or (**Inverse**) when one variable increases as the other one decreases, and **No Correlation** if there is not relation between changes in the two variables. Table 3 reports the scale of the values with which to evaluate the strength of the correlation results.

A typical analysis workflow using COVID-WAREHOUSE comprises the following steps (which are performed automatically by COVID-WAREHOUSE, see Figure 3):Data Collection: COVID-WAREHOUSE read from the local repository the climate, air pollution, and Italian COVID-19 data, importing data automatically into the Data-Container.Data Cleaning: data are automatically cleaned and transformed by means of automatic ETL approaches, tailored for COVID-19, air pollution and climatic data. In particular, we implemented several ETL methods based on Regular Expression (RE) to extract, clean and format attributes. For each data set we figure out the main attributes e.g., key-attributes including Region, Province, and Date-Time, making it possible to relate all the downloaded data. Region and Province attributes have been cleaned removing or replacing special characters, e.g., due to the use of *Latin-1* encoding and converted in *UTF-8* encoding, avoiding in this way possible mismatches. Attribute Date-Time has been cleaned, encoded in **UTF-8**, and converted in a new and common format e.g., “*YYYY/MM/DD*”. All the other attributes have been cleaned by removing or replacing special characters and encoded in *UTF-8*. As an example, the rain attribute present in the climatic data set contains literal values e.g., “*sereno, pioggia, etc.*”, that need to be converted into numerical values. Thus, the rain attribute has been discretized by mapping specific climatic conditions using 4 values e.g., 0 indicates absence of rain, 1 represents all the types of rain, 2 refers to sleet, and, finally, 3 indicates snow.Data Merging: data are automatically merged by using customized joining schema, through the joining and merging functions available in Python Pandas Data Frames, making it possible to obtain a single ReData Analysisconciled Table from the all the input data sets. Cleaned Data provides the foundation to create the Reconciled Table, because now all the key attributes are in the same format and encoding, making it possible to represent multidimensional concepts in a more efficient way.Data Aggregation: it is implemented by using the functions available in Pandas Data Frames. In this way, it is possible to yield a condensed version of the DW called *Data Mart* obtained from the Reconciled Table. Data Marts make it possible to quickly aggregate data because they are small in size with respect to the overall DW, and are more flexible to yield multidimensional cubes. Since, DW has data coming in from multiple data sources, Data Marts help to efficiently organize all of data in a multidimensional format (cube) enabling to perform data analysis in a straightforward and more efficient way.Data Analysis and Visualization: it allows to perform statistical analysis and visualization on the predefined cubes. In particular, COVID-WAREHOUSE provides the users with an automatic tools to perform correlation analysis on data cubes, as well as data report. To make data analysis more straightforward, correlation results are displayed using a Heatmap representation, making it easier to assess the correlation strength between two variables.

To figure out the possible correlation among environmental factors (i.e., air pollution, and climatic conditions) that can contribute to the spread of the COVID-19 disease (i.e., the total number of positive cases recorded daily), we aggregated data in the following multidimensional cubes:**[CUBE-1]:** The total number of positive cases recorded in the Lombardia region has been represented in a 3D space, whose dimensions are the recording date, the provinces of Lombardia, the meteorological information w.r.t. average wind speed (Km/h) and the detected air pollution PM10 (μg/m${}^{3}$), and PM2.5 (μg/m${}^{3}$) values, and the considered measure is the number of positive people.**[CUBE-2]:** The total number of positive cases recorded in the Lombardy region has been represented in a 3D space, whose dimensions are the recording date, the provinces of Lombardy, the climatic conditions (presence/absence of rain) and the detected air pollution PM10 (μg/m${}^{3}$), and PM2.5 (μg/m${}^{3}$) values, and the considered measure is the number of positive people.**[CUBE-3]:** The total number of positive cases recorded in the Puglia region has been represented in a 3D space, whose dimensions are the recording date, the provinces of Puglia, the climatic conditions wind (Km/h) and the detected air pollution PM10 (μg/m${}^{3}$) values, and the measure is the number of positive people.**[CUBE-4]:** The total number of positive cases recorded in the Puglia region has been represented in a 3D space, whose dimensions are the recording date, the provinces of Puglia, the climatic conditions (presence/absence of rain) and the detected air pollution PM10 (μg/m${}^{3}$) values, and as measure the number of positive people has been taken into account.**[CUBE-5]:** The total number of positive cases recorded in the whole Italian territory has been represented in a 3D space, whose dimensions are the recording date, all Italian location regions, the climatic conditions presence/absence of rain, wind (Km/h), and as measure the number of positive people has been considered.

To compute correlation for each one of the 5 yielded cubes, it was needed to identify the independent and dependent variables. In all cubes, the dependent variable is the number of positive cases, for *CUBES-[1,2,3,4]* the independent variables are air pollution and climatic conditions, whereas for *CUBE-5* the independent variable is the climatic conditions (we do not have pollution data for all Italy). To calculate the correlation between more than two variables, we used the multiple correlation coefficient [25]. The coefficient of multiple correlation generalizes the standard coefficient of correlation. It is used in multiple regression analysis to assess the quality of the prediction of the dependent variable. The coefficient of multiple correlation is defined in Equation (Equation 1):(1)$$Corr=\sqrt{\frac{xy^{2}+yz^{2}-2xz*yz*xy}{1-xy^{2}}}$$

In Equation (Equation 1), *x* and *y* are the independent variables and *z* is the dependent variable, on which to map the dependent and independent variables of the analyzed cubes. We calculate the multiple correlation for all the provided cubes by using Equation (Equation 1). To make correlation analysis easier by using the scale of correlation values summarized in Table 3, multiple correlation results are visualized as heatmaps. Figure 4 shows the correlation values calculated from CUBE-1. It is worthy to note that, the dimension *wind* appears to be strongly positively correlated with the number of positive cases (corr=0.8), whereas particulate particles PM10 present a strong correlation with the number of positive cases (corr=0.7). This result, should pinpointing out that faster wind speed, along with he presence of higher level of particulate particles in the air, might contribute to increase the spread of the COVID-19.

Figure 5 shows the correlation values computed from CUBE-2. The heatmap highlights a strong negative correlation (corr = −0.9) between rain and the number of positive cases. This should mean that, when it is raining, the number of infections should decrease. *PM10* presents both a strong correlation (corr = 0.7), with respect to the number of positive cases. *PM2.5* presents both a strong negative correlation (corr = −0.7) with the number of positive cases. The correlation for both particulate particles, is due to the correlation with the rain, showing a strong negative correlation (corr = −0.9), meaning that, when there is rain the particulate level in the air tends to drop down, limiting the spreading of the virus.

Figure 6 shows the correlation values computed from CUBE-3. PM10 presents a strong correlation (corr = 0.9) with the total number of positive cases. Wind has a strong correlation (corr = 0.9) with the total number of positive cases. This result shows that even for the Puglia region as well as for Lombardia, wind and high values of particulate in the air could contribute to the spread of the virus.

Figure 7 shows the correlation values computed from CUBE-4. PM10 presents a strong, positive correlation (corr = 0.9) with the total number of positive cases. Rain has a strong negative correlation (corr = −0.7) with the total number of positive cases. As well as for CUBE-2, even for CUBE-4 rain and particulate particles have a strong negative correlation (corr = −0.8).

Figure 8 shows the correlation values calculated from CUBE-5. The attribute rain presents a positive strong correlation (corr = 0.7) with the total positive cases, whereas wind presents a moderate negative correlation (corr = −0.6) with the total positive cases.

Our findings indicate that COVID-19 pandemic is significantly spread in regions characterized by high concentration of particulate in the air and the absence of rain and wind, as even stated in several scientific paper investigating the climatic and pollution effect on the COVID-19 spreading in other countries [13,15,16].

## 6. Conclusions

In this paper, we proposed a new approach for building a data warehouse from a limited volume of data, such as Italian COVID-19 data. We developed an automatic methodology to enrich Italian COVID-19 data with air pollution and climatic data in the Italian regions. Additionally, to the heterogeneous nature of the data sources participating in the construction of our data warehouse model, we defined some merging models to handle, clarify (eliminate ambiguity) and unify data in a reconciled model. Merged models are stored and available in the data warehouse repository. The presence of these models may contribute to reduce the complexity of other phases of the data warehouse life cycle (optimization, personalization, and evolution management). Indeed, we implemented a set of ETL methods that are exploiting the available merging models, can automatize the whole data analysis workflow, starting from the data import up to the data visualization and analysis.

The proposed approach is implemented in a software framework called COVID-WAREHOUSE, fully developed in Python. COVID-WAREHOUSE allows users to import, clean, merge, and aggregate heterogeneous data, to provide the central repository (data warehouse) from which to assemble multidimensional cubes. In the current version, COVID-WAREHOUSE supports users to enrich COVID-19 data with air pollution and climatic data, making it possible to create multidimensional data cubes (by using a different set of measures and values). In this way, COVID-WAREHOUSE allows to represent data along with some measure of interest (not available in the standard COVID-19 data), helping to get a border scenario by establishing trends from a variety of viewpoints aggregated data. Thus, multidimensional analysis of data helps to capture and quantify significant changes in aggregated COVID-19 data, bringing to light aspects of these changes that can help to explain the underlying events and mechanisms which drive them.

To facilitate the analysis of the concerned multidimensional data cubes, the COVID-WAREHOUSE allows the calculation of the statistical correlation among multiple variables. The obtained correlation results are presented as heat-maps, making it easier to understand if there is a relationship between the investigated variables.

Our preliminary analysis of COVID-19 data integrated with air pollution and climate data, have shown that in Lombardia and Puglia regions the levels of PM10, and the average wind speed show a positive strong correlation with the number of positive cases, with the correlation coefficients ranging from 0.7 to 0.9. On the other hand, in both Lombardia and Puglia regions the level of PM10, and rain show a strong negative correlation with the number of positive cases, with the correlation coefficients ranging from 0.7 to 0.9. However, by analyzing the climate data for all the Italian regions, the absence of rain shows a strong positive correlation with the number of positive cases, and the correlation coefficient equals to 0.7, while the average wind speed shows a moderate negative correlation with the number of positive cases, with a correlation coefficient equals to −0.6.

Our preliminary findings could suggest that COVID-19 pandemic has significantly spread in regions characterized by high concentration of particulate in the air and absence of rain and wind, according with similar scientific results in the literature.

As future work, we are planning to improve the data cubes creation by adding a graphical user interface, supporting users to create data cubes through drag-and-drop functions. Also, we are extending ETL functions and data warehouse models to handle COVID-19 data from other countries (as they become available), to make multiple data comparison straightforward. We are also extending the statistical analysis library available in COVID-WAREHOUSE, providing users with both an analysis and data storage environment, avoiding the need to use additional software for data analysis.

## Figures and Tables

**Figure 1 ijerph-17-05596-f001:**
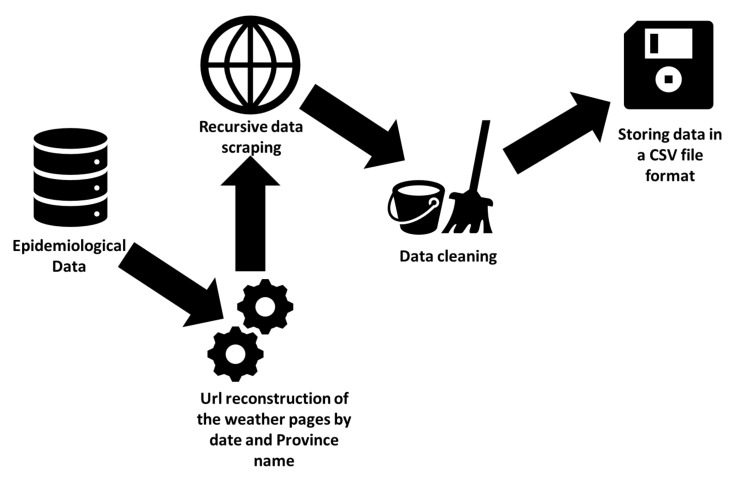
Weather data collection pipeline. A list of dates and Province names are extracted from the epidemiological data previously described. Then, the extracted province names are combined with all dates to reconstruct the URL of each page that contains relative information in the meteorological website. Data is extracted through data scraping, then raw data is normalized (e.g., by fixing inconsistency, providing uniformity of unit measurement for each date and province) and cleaned (e.g duplicate instances and redundant information removal, one-hot encoding of meteorological phenomena). Finally data is stored in a CSV file format.

**Figure 2 ijerph-17-05596-f002:**
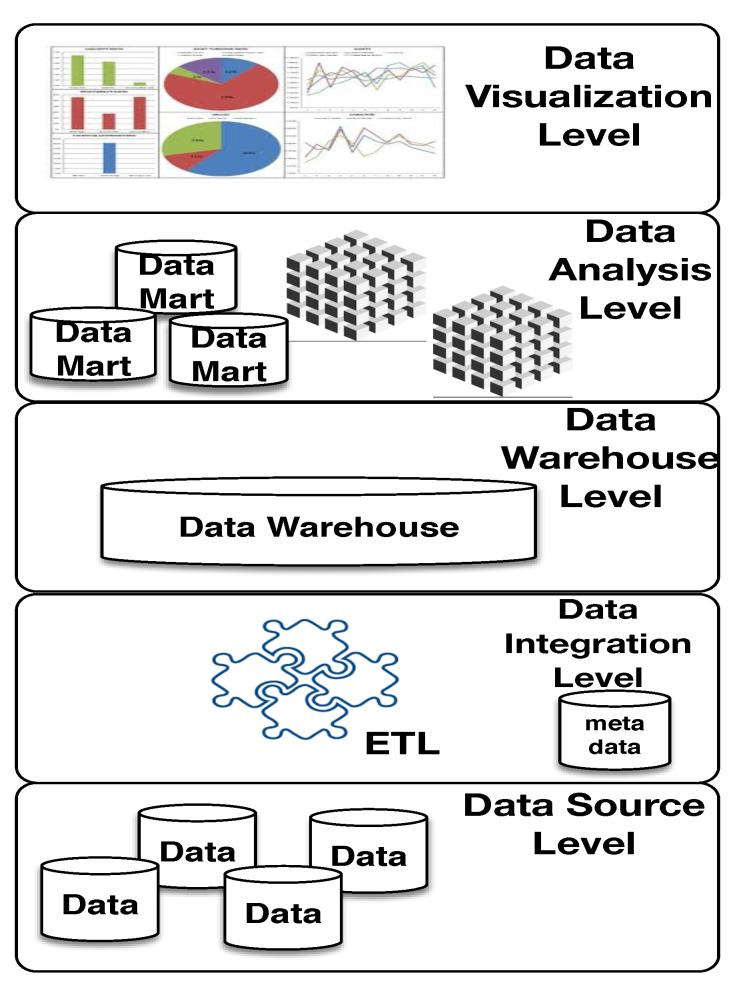
The COVID-WAREHOUSE architecture, implemented as 5 independent and cooperating levels.

**Figure 3 ijerph-17-05596-f003:**
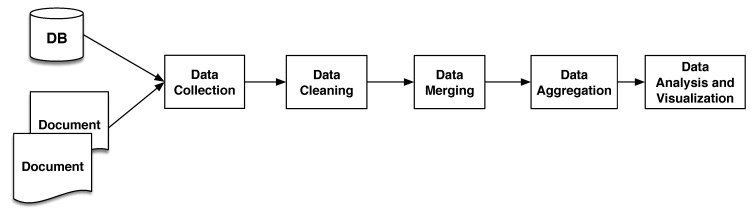
The main modules of COVID-WAREHOUSE analysis pipeline. **Data Collection** allows to store the downloaded raw data locally. **Data Cleaning** provides ETL methods based on RE, enabling the cleaning, replacing and deletion of special characters present into the raw data, that can compromise the next steps and analysis. **Data Merging** uses cleaned data to produce the Reconciled Table. **Data Aggregation** yields a condensed version of the DW called *Data Mart* obtained from the Reconciled Table, from which to obtain the multidimensional cubes. **Data Analysis and Visualization** allows to perform statistical analysis and Heatmap visualization on the predefined cubes.

**Figure 4 ijerph-17-05596-f004:**
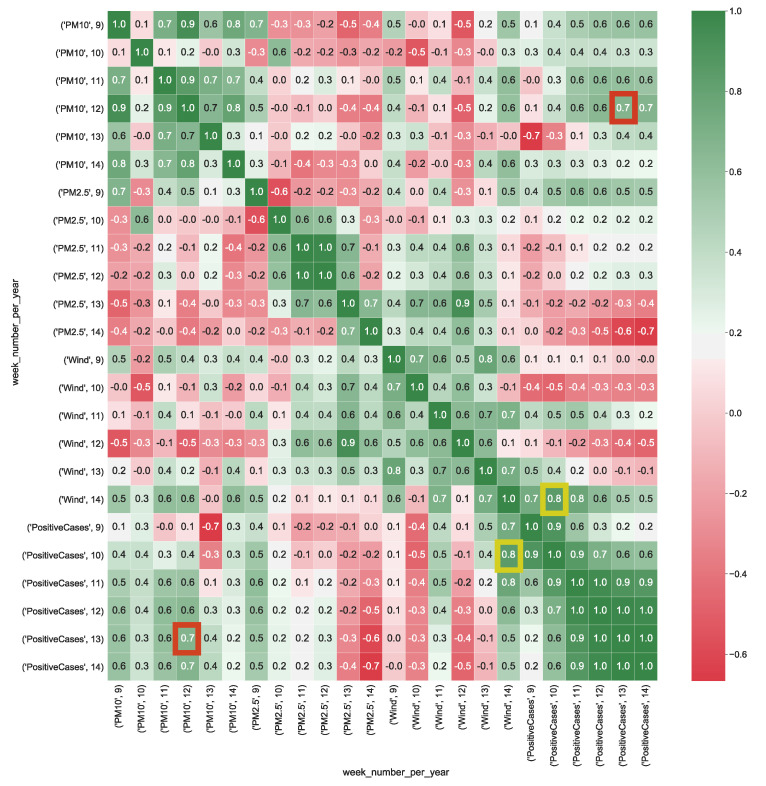
Heatmap represents the correlation between aggregated COVID-19 data with air pollution and wind data (km/h) detected in the Lombardia region. The heatmap’s labels refer to the attribute’s measured value in a specific week of the year. For instance, *’PM10’, 9* refers to the level of PM10 (μg/m${}^{3}$) in the air measured in the *9-th* week of the year. In Figure, the yellow squares highlight the strong correlation between positive cases and the presence of wind, whereas red squares show the strong correlation between air particulate and the number of positive cases.

**Figure 5 ijerph-17-05596-f005:**
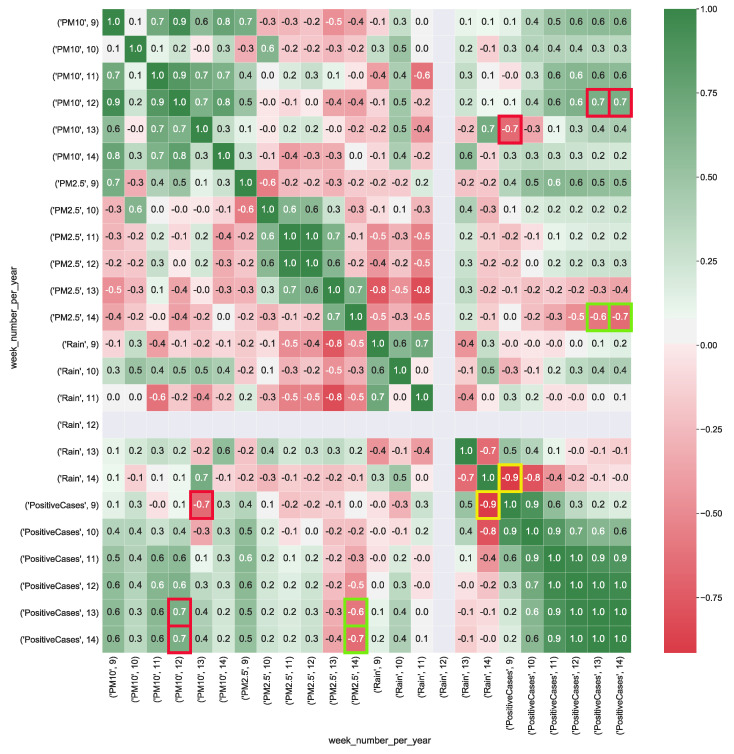
Heatmap representation of the correlation between COVID-19 data aggregate with the air pollution and rain data (boolean variable) detected in the Lombardia region. The heatmap’s labels refer to the attribute’s measured value in a specific week of the year. For instance, *’PM10’, 9* refers to the level of PM10 (μg/m${}^{3}$) in the air measured in the *9-th* week of the year. In Figure, the yellow squares highlight the strong negative correlation between positive cases and the presence of rain, whereas red squares show both strong positive correlation between the PM10 air particulate and the number of positive cases. Green squares show strong negative correlation between the PM2.5 air particulate and the number of positive cases.

**Figure 6 ijerph-17-05596-f006:**
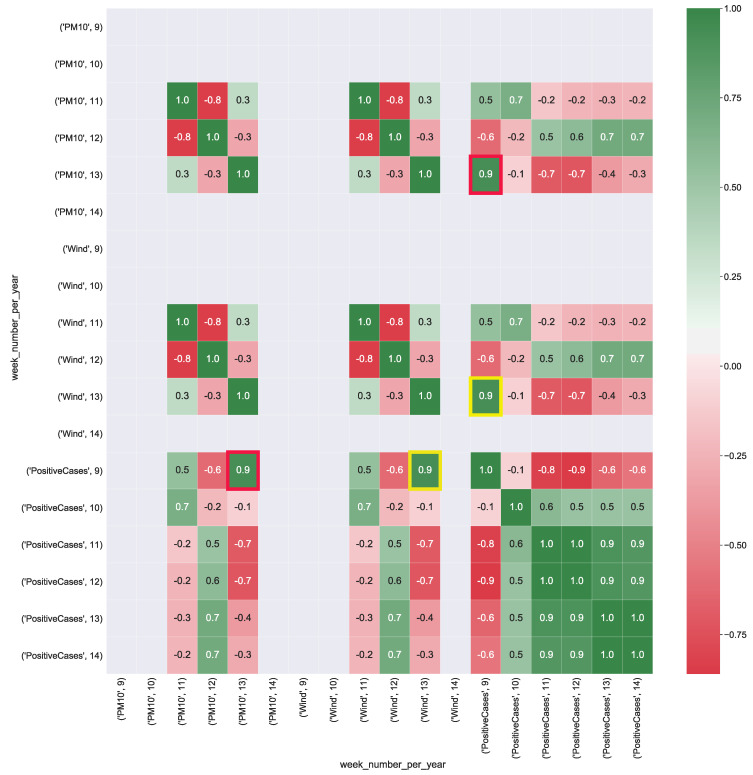
Heatmap representation of the correlation between COVID-19 data aggregate with the air pollution and wind data (km/h) measured in the Puglia region. The heatmap’s labels refer to the attribute’s measured value in a specific week of the year. For instance, *’PM10’, 9* refers to the level of PM10 (μg/m${}^{3}$) in the air measured in the *9-th* week of the year. In Figure the yellow squares highlight the strong correlation between positive cases and the presence of wind, whereas red squares show the strong correlation between air particulate and the number of positive cases.

**Figure 7 ijerph-17-05596-f007:**
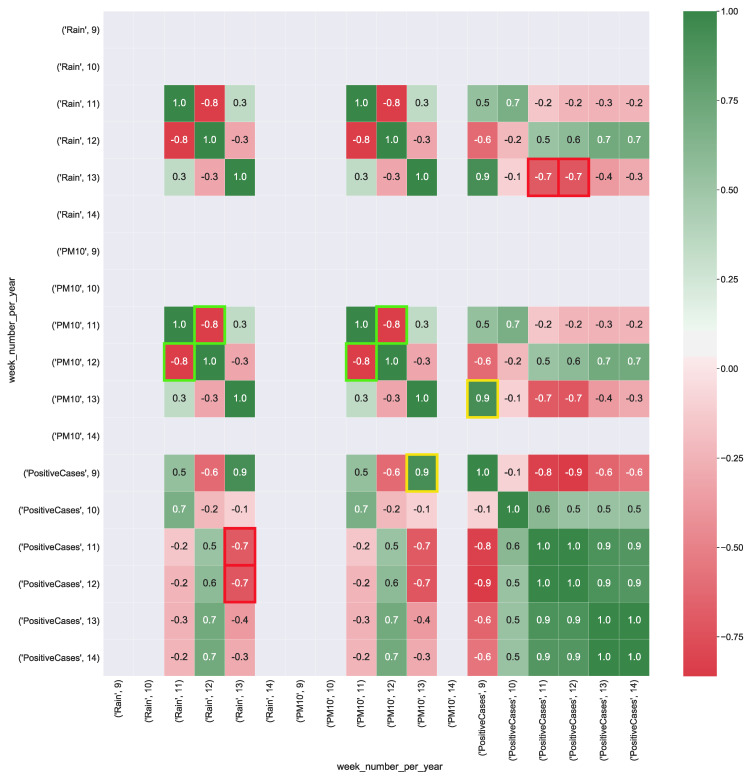
Heatmap representation of correlation between COVID-19 data aggregate with the air pollution and rain data (boolean variable) detected in the Puglia region. The heatmap’s labels refer to the attribute’s measured value in a specific week of the year. For instance, *’PM10’, 9* refers to the level of PM10 (μg/m${}^{3}$) in the air measured in the *9-th* week of the year. In Figure, the yellow squares highlight the strong positive correlation between positive cases and the presence of air particulate PM10, whereas red squares show the strong negative correlation between rain and the number of positive cases. Green squares show strong negative correlation between rain and PM10 particulate.

**Figure 8 ijerph-17-05596-f008:**
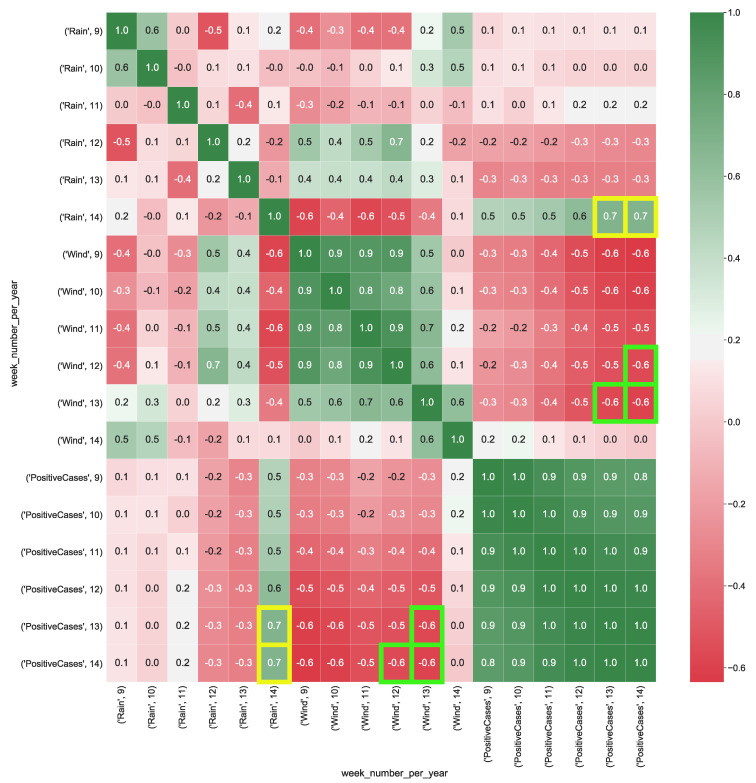
Heatmap representation of the data cube aggregating together the meteorological data for all the Italian regions and the COVID-19 data. The heatmap’s labels refer to the attribute’s measured value in a specific week of the year. For instance, *’PM10’, 9* refers to the level of PM10 (μg/m${}^{3}$) in the air measured in the *9-th* week of the year. In Figure the yellow squares highlight the strong correlation between positive cases and the presence of rain, whereas green squares show the strong negative correlation between wind and the number of positive cases.

**Table 1 ijerph-17-05596-t001:** The detected PM10 values for the city of Bergamo (Via Meucci Station), available on the ARPALombarida web site. The measures start from 2020/01/01, up to the date of writing. The −999 value means a missing value.

Date/Time	PM10-μg/m${}^{3}$
2020/01/01 00:00	42
2020/01/02 00:00	52
2020/01/03 00:00	49

**Table 2 ijerph-17-05596-t002:** The detected PM2.5 values for the city of Bari, available on the ARPAPuglia web site. The measures start from 2020/02/24, up to the date of writing.

DetectionUnit	Date/Time	Prov	CityHall	Particulate	Values	Warn
Bari-Cavour	2020-02-24 00:00:00.0	Bari	Bari	PM2.5	23	
Bari-Cavour	2020-02-25 00:00:00.0	Bari	Bari	PM2.5	24	
Bari-Cavour	2020-02-26 00:00:00.0	Bari	Bari	PM2.5	13	

**Table 3 ijerph-17-05596-t003:** The scale of correlation strength values.

Correlation Value	Strength
−1.0 ≤ *C* ≤ −0.7	Strong Negative Correlation
−0.7 < *C* ≤ −0.3	Moderate Negative Correlation
−0.3 < *C* ≤ 0.0	Weak Negative Correlation
0.0 < *C* ≤ 0.3	Weak Positive Correlation
0.3 < *C* ≤ 0.7	Moderate Positive Correlation
0.7 < *C* ≤ 1.0	Strong Positive Correlation

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
