# Peer review of "COVID-WAREHOUSE: A Data Warehouse of Italian COVID-19, Pollution, and Climate Data"

_ijerph, 2020, doi:10.3390/ijerph17155596_

Round 1
Reviewer 1 Report
Authors present a data warehouse (DW) integrating COVID-19 data with air pollution and climatic data in the Italian regions and apply it to some real case studies, based on real data collected in Italy. The system has been fully implemented in a software framework called COVID-WAREHOUSE, supporting OLAP analysis and allowing to compute the correlation between the data stored in the data warehouse, such as COVID-19 measures (e.g. cases, deaths, etc.), and climate and pollution data. The paper is interesting, well written, and technically sound. The authors introduce the problem and then describe the framework highlighting its capabilities. I think the content would be interesting for readers. From the presented case studies, it seems that this research can help public decision-makers to find correlation between contaminants of pollution and weather conditions, with the spread of COVID-19 pandemic, in each of the Italian region. The authors should therefore also address the following minor comments, in order to improve the presentation's quality:
- The introduction could be improved including recent contributions in the body of literature.
- The related work could be improved adding some classical framework for data collection, harmonization, and integration.
- A more explanation of how the processes of normalization/denormalize work in the data warehouse, would be very helpful for readers.
- There are some typos, here are reported some examples: Abstract line 12: “data fur further” must be “data for further”Section 1. line 82: “allows to study possible correlation” must be “allows to study the possible” correlation” Section 1. line 86: “have been integrated in the data warehouse” must be “have been integrated into the data warehouse” Section 3. Line 174 “often in a unstructured “ must be “often in an unstructured” Section 4. Line 268 “to obtain the new the DFM” must be “to obtain the new DFM” Section 4. Line 303 “to represent spreadsheet or SQL table.” Must be “to represent a spreadsheet or SQL table.”
Author Response
Reviewer 1
Comments and Suggestions for Authors
Authors present a data warehouse (DW) integrating COVID-19 data with air pollution and climatic data in the Italian regions and apply it to some real case studies, based on real data collected in Italy. The system has been fully implemented in a software framework called COVID-WAREHOUSE, supporting OLAP analysis and allowing to compute the correlation between the data stored in the data warehouse, such as COVID-19 measures (e.g. cases, deaths, etc.), and climate and pollution data. The paper is interesting, well written, and technically sound. The authors introduce the problem and then describe the framework highlighting its capabilities. I think the content would be interesting for readers. From the presented case studies, it seems that this research can help public decision-makers to find correlation between contaminants of pollution and weather conditions, with the spread of COVID-19 pandemic, in each of the Italian region. The authors should therefore also address the following minor comments, in order to improve the presentation's quality:
- The introduction could be improved including recent contributions in the body of literature.
Authors: We improved Introduction by reporting some recent contributions.
In particular we added the following sentences colored in red in the text:
SARS-CoV-2 has spread over all the world in less than six months, causing more than 10 million tested-positive cases and more than half a million confirmed deaths \cite{world2020coronavirus}
A global response has been quickly developed in the form of collective data collection and analysis efforts, which are generally aimed to understand SARS-CoV-2 biology and delivering therapeutic solutions in clinical/pharmacological protocols \cite{le2020covid}.
- The related work could be improved adding some classical framework for data collection, harmonization, and integration.
Authors: We extended and improved the Related Work section. In particular we organized it into two subsections: the novel Subsection 2.1 that describes some of the available open source frameworks for building data warehouses; and the Subsection 2.2 that comprises the literature works describing the joint analysis of COVID-19, climate and pollution data, already present in the previous version of the manuscript.
- A more explanation of how the processes of normalization/denormalize work in the data warehouse, would be very helpful for readers.
Authors: We added subsection 4.1 in Section 4 discussing how the processes of normalization and denormalization work respectively in the database and data warehouse.
- There are some typos, here are reported some examples: Abstract line 12: “data fur further” must be “data for further”Section 1. line 82: “allows to study possible correlation” must be “allows to study the possible” correlation” Section 1. line 86: “have been integrated in the data warehouse” must be “have been integrated into the data warehouse” Section 3. Line 174 “often in a unstructured “ must be “often in an unstructured” Section 4. Line 268 “to obtain the new the DFM” must be “to obtain the new DFM” Section 4. Line 303 “to represent spreadsheet or SQL table.” Must be “to represent a spreadsheet or SQL table.”
Authors: We thank the Reviewer for pointing out these errors. We fixed all typos and improved English with the help of a native English speaker.
Reviewer 2 Report
The article "COVID-WAREHOUSE: A Data Warehouse of Italian COVID-19, pollution, and climate data" presents a proposal to design and develop a data warehouse about COVID-19 (COVID-WAREHOUSE) from a variety of data sources, and to correlate to pollution and climate data. The primary motivation is to have a free and public multidimensional data source about COVID-19 to support data analysis. In general, COVID-WAREHOUSE creation follows the traditional steps of data warehouse creation. The data views are essential as data scenarios, and these scenarios were supported by a heatmap visualization representing the attributes' statistical correlation.
- the manuscript needs an english review, mainly in the first part of it
- it is no clear for me if the data warehouse is available to free access or if there is a cloud service available? How can I create another thematic data warehouse?
-I believe that there are not many details to reproduce the work, for instance, strategies to clean and integrate data, etc. However, the overview is good. The steps show small samples of dataset transforming and integrate with other data sources.
- Explore more Figure 4
- add some description of how to read the labels of the Figures 5,6,7,8,9, for instance 'Wind', 11 or 'Wind '12'
- the main weaknesses of the article are not to compare with other proposals to create a data warehouse, the related works are focused only on covid-19 data, and important development details are missing
minor revisions
Minor revisions (sugestions)
Line 25 suppress the comma
Line 28 fully --> full
Line 33 modelling --> modeling
Line 49 entity ïƒ entities
Line 52 dowloaded -->downloaded
Line 67 suppress the comma, or change the place
Line 83 follow --> follows
Line 93 reports --> report
Line 97 PM is not defined, to --> for
Line 174 a unstructured --> an unstructured
Line 176 e other occurrences Hospitalised ïƒ Hospitalized
Figure 1 horizontal line layout
Detail in the text the type of clean data and transformation process was applied
Figure 2 to invert the direction of process
326 paragraph format
I consider that figure 3 is not so useful, I miss a process figure about how someone can create the data marts from row data, how to connect the tables, cleaning process, etc
Between lines 369 and 378, the text informs that the data correlation analysis is in the range [-1,1], but table 3 considers only positive range, add some information about that
Figure 4, you can highlight some techniques or tools applied in each step
Line 402, 406, 410, 414 417 measure is ?
In page 13, without number line, add a reference about “the multiple correlation Coeficiente”
In Figure 5, Figure 6, Figure 7, Figure 8 and Figure 9 highlight the findings in the visualization, for instance, using a green square
It is more natural use red color for negative values and blue color or green for positive values
Author Response
Reviewer 2
Comments and Suggestions for Authors
The article "COVID-WAREHOUSE: A Data Warehouse of Italian COVID-19, pollution, and climate data" presents a proposal to design and develop a data warehouse about COVID-19 (COVID-WAREHOUSE) from a variety of data sources, and to correlate to pollution and climate data. The primary motivation is to have a free and public multidimensional data source about COVID-19 to support data analysis. In general, COVID-WAREHOUSE creation follows the traditional steps of data warehouse creation. The data views are essential as data scenarios, and these scenarios were supported by a heatmap visualization representing the attributes' statistical correlation.
- the manuscript needs an english review, mainly in the first part of it
Authors: We fixed several typos and improved English with the help of a native English speaker.
- it is no clear for me if the data warehouse is available to free access or if there is a cloud service available? How can I create another thematic data warehouse?
Authors: In the current version, COVID-WAREHOUSE can only be used locally from the command line, and it is not accessible through the network via a web interface. To make COVID-WAREHOUSE available online, we plan to implement a new version of the system by implementing its GUI and its web architecture, making it possible to use all the functionalities available in COVID-WAREHOUSE by means a common web-browser and in a secure way.
-I believe that there are not many details to reproduce the work, for instance, strategies to clean and integrate data, etc. However, the overview is good. The steps show small samples of dataset transforming and integrate with other data sources.
Authors: We have extended and better described the main steps of the analysis workflow of the DW in Section 5. We added the following paragraphs (colored in red in the text):
“2. Data Cleaning: data are automatically cleaned and transformed by means of automatic ETL approaches, tailored for COVID-19, air pollution and climatic data. In particular, we implemented several ETL methods based on Regular Expression (RE) to extract, clean and format attributes. For each data sets we figure out the main attributes e.g., key-attributes including Region, Province, and Date-Time, making it possible to relate all the downloaded data. Region and Province attributes have been cleaned removing or replacing special characters i.e., due to the use of \textit{Latin-1} encoding and converted in \textit{UTF-8} encoding, avoiding in this way possible mismatches. Attribute Date-Time has been cleaned, encoded in \textbf{UTF-8}, and converted in a new and common format e.g., "\textit{YYYY/MM/DD}". All the other attributes have been cleaned removing or replacing special characters and encoded in \textit{UTF-8}. As an example, the rain attribute present in the climatic data set contains literal values e.g., "\textit{sereno, pioggia, etc.}", that need to be converted into numerical values. Thus, the rain attribute has been discretized by mapping specific climatic conditions using 4 values e.g., 0 indicates absence of rain, 1 represents all the types of rain, 2 refers to sleet, and finally 3 indicates snow.
- Data Merging: data are automatically merged by using customized joining schema, through the joining and merging functions available in Python Pandas Data Frames, making it possible to obtain a single reconciled table from the all the input data sets, making it possible to obtain a single reconciliated table from the join of all the input data sets. Cleaned Data provides the foundation to create the reconciliated Table, because now all the key attribute are in the same format and encoding, making it possible to represent multidimensional concepts in a more efficient way.
- Data Aggregation: Data aggregation is implemented by using the functions available in Pandas Data Frames. In this way, it is possible to yield condensed versions of DW called \texiData{Data Marts} obtained from the Reconciled Table. Data Marts make it possible to quickly aggregate data because they are small in size with respect to the overall DW, and are more flexible to yield multidimensional cubes. Since, the DW collects data coming in from multiple data sources, Data Marts help to efficiently organize all of data in a multi-dimensional format (cube) enabling to perform data analysis in a straightforward and more efficient way.”
- Explore more Figure 4
Authors: We extended the Figure 4 caption by summarizing the functions of each module of the analysis pipeline. Please note that now this figure is the number 3.
- add some description of how to read the labels of the Figures 5,6,7,8,9, for instance 'Wind', 11 or 'Wind '12'
Authors: We added in the Figures’ caption a detailed description of the labels meaning.
- the main weaknesses of the article are not to compare with other proposals to create a data warehouse, the related works are focused only on covid-19 data, and important development details are missing
Authors: We thank the Reviewer for pointing out this. We want to specify that our project is specifically devoted to the automatic building of a data warehouse based on publicly available CSV data and especially COVID-19 data, that are often not curated, and this is one main reason because we do not cite many proposal for building general purpose data warehouses. Anyway, some related work is now reported in the new subsection 2.1.
In particular, both COVID-19, climate and pollution data are released on a time based way in csv files that need to be cleaned and preprocessed before their ingestion. The only one system that allows to automatically build a data warehouse starting from csv files and COVID-19 csv files is “The Stitch COVID-19 Data” initiative available at (https://www.stitchdata.com/integrations/covid-19/), but unfortunately it is not for free, so we were not able to fully test that system to make a comparison with ours.
minor revisions
Minor revisions (sugestions)
Line 25 suppress the comma
Line 28 fully --> full
Line 33 modelling --> modeling
Line 49 entity ◊ entities
Line 52 dowloaded -->downloaded
Line 67 suppress the comma, or change the place
Line 83 follow --> follows
Line 93 reports --> report
Line 97 PM is not defined, to --> for
Line 174 a unstructured --> an unstructured
Line 176 e other occurrences Hospitalised ◊ Hospitalized
Authors: We fixed all these typos
Figure 1 horizontal line layout
Detail in the text the type of clean data and transformation process was applied
Authors: We updated the Figure1 using a horizontal layout as suggested by the reviewer.
Figure 2 to invert the direction of process
Authors: We thanks the reviewer to pointing out this lack; we put the process in the correct order.
326 paragraph format
Authors: We fixed this typo.
I consider that figure 3 is not so useful, I miss a process figure about how someone can create the data marts from row data, how to connect the tables, cleaning process, etc
Authors: We removed Figure 3. Moreover, to better explain the main steps of data mart creation starting from raw data, we extended the caption of the former Figure 4, that now is Figure 3 having deleted old Figure 3.
Between lines 369 and 378, the text informs that the data correlation analysis is in the range [-1,1], but table 3 considers only positive range, add some information about that
Authors: We added to the Table 3 the missing information related to the negative correlation.
Figure 4, you can highlight some techniques or tools applied in each step
Authors: We have extended and better described the main steps of the analysis workflow by extending the caption of Figure 4 (now Figure 3).
Line 402, 406, 410, 414 417 measure is ?
Authors: we thank the reviewer to pin pointing out this typo. We added to each attribute the measurement units.
In page 13, without number line, add a reference about “the multiple correlation Coeficiente”
Authors: We added the reference to the multiple correlation coefficient.
In Figure 5, Figure 6, Figure 7, Figure 8 and Figure 9 highlight the findings in the visualization, for instance, using a green square
Authors: In Figure 5, Figure 6, Figure 7, Figure 8 and Figure 9 we highlight the findings using different colored squares.
It is more natural use red color for negative values and blue color or green for positive values
Authors: We agree with the Reviewer and we produced new heatmaps using red and green colors.